# The oral intake of specific Bioactive Collagen Peptides (BCP) improves gait and quality of life in canine osteoarthritis patients—A translational large animal model for a nutritional therapy option

**Britta Dobenecker**[1]*, **Linda Franziska Böswald**[1], **Sven Reese**[1], **Stephanie Steigmeier-Raith**[2], **Lukas Trillig**[2], **Steffen Oesser**[3], **Michael Schunck**[3], **Andrea Meyer-Lindenberg**[2], **Jutta Hugenberg**[4]

**1** Department of Veterinary Science, Ludwig-Maximilians-Universität München, München, Germany, **2** Clinic for Small Animal Surgery and Reproduction, Ludwig-Maximilians-Universität München, München, Germany, **3** Collagen Research Institute, Kiel, Germany, **4** Gelita AG, Ebersbach, Germany

* Dobenecker@lmu.de

## Abstract

### Objective

Osteoarthritis (OA) is the most common joint disorder in humans and dogs. Due to its chronic progressive nature, the predominant clinical signs after a certain point are pain and immobility. The similar pathogenesis allows conclusions to be drawn from canine to human OA. Current treatments are limited and often attempt to treat OA symptoms rather than improve joint structure and function. Collagen hydrolysates as oral supplements are a promising therapeutic option to achieve this advanced therapeutic aim in both species. The effects of oral supplementation were therefore investigated in canine OA patients.

### Method

In a systematic, placebo-controlled, double-blind interventional study in 31 dogs with naturally occurring OA, the efficacy of oral supplementation of specific bioactive collagen peptides (BCP) was tested in comparison to the approved combination of the active substances omega-3 fatty acids and vitamin E. The dogs were examined on a horizontal treadmill with 4 integrated piezoelectric force plates at the beginning and end of a twelve-week test period. At both points, the owners completed a specific questionnaire containing the validated Canine Brief Pain Inventory (CBPI) and the dogs were fitted with accelerometers to record total daily activity data.

### Results

Only the oral supplementation of BCP resulted in a significant improvement of several kinetic parameters measured using a force-plate fitted treadmill, and the quality of life assessed by CBPI, while accelerometry was unaffected by the intervention.

**Data Availability Statement:** All relevant data are within the paper and its Supporting Information files.

**Funding:** GELITA AG, Germany, provided support in the form of financial compensation for authors (B. Dobenecker, L. Böswald, S. Reese), but did not have any additional role in the study design, data collection and analysis, decision to publish, or preparation of the manuscript.

**Competing interests:** Jutta Hugenberg is employed by GELITA AG, Eberbach, Germany, which funded the study. Steffen Oesser is co-inventor of patents concerning the use of collagen peptides. This does not alter the adherence to PLOS ONE policies on sharing data and materials.

## Conclusion

The results of this three-month BCP supplementation study using objective measurement parameters in dogs with naturally occurring OA demonstrate an efficacy, suggesting the therapeutic use of BCP in canine OA patients and demonstrating the relevance of this collagen hydrolysate formulation for the treatment of OA in human patients as well.

## Introduction

In human medicine, osteoarthritis is a joint disease with increasing global prevalence, especially among the elderly [1]. The use of translational animal models is a valuable strategy in the process of developing therapy options. Companion animals like dogs provide a valid model for spontaneously occurring joint OA and pain medication [2], so the results of studies in these species can advance both veterinary and human medicine. Due to the anatomical and pathological commonalities between dogs and humans, spontaneously occurring osteoarthritis (OA) in dogs is a frequently used model in translational medicine for human OA [3–5]. Spontaneous OA pathology in dogs and humans is largely similar, and the canine model is good for evaluating the effectiveness of different treatment options [6].

In veterinary medicine, it is known that a very high number of dogs suffer from OA, a chronic progressive disease that usually occurs in older age and has a prevalence of up to 80% in dogs older than eight years, with overweight and obesity increasing the risk of the disease [7, 8]. Although the disease can occur in younger dogs, it is usually diagnosed when it reaches a more advanced stage with increasingly obvious clinical signs such as stiffness, lameness, and unwillingness to exercise, jump or climb [9]. OA is characterized by loss of articular cartilage in the synovial joints, subchondral bone sclerosis, synovitis, and thickening of the capsule. The disease can be defined as a painful condition that affects the animal's well-being.

Non-surgical treatment options for OA are currently limited mostly to pain management, which can alleviate symptoms but not act causative. Non-steroidal anti-inflammatory drugs (NSAIDs) are frequently used for their anti-inflammatory and analgesic effects in OA, despite the incidence of adverse effects such as gastrointestinal problems, renal insufficiency, anorexia, lethargy and even death [10]. Prolonged administration of NSAIDs in older dogs, which OA patients usually are, is not recommended [11]. In a systematic literature review, Innes et al. (2010) evaluated information on the efficacy and safety of long-term use of NSAIDs in dogs to treat OA [12]. They concluded that long-term use is associated with reduced safety, although there is a lack of robust data in this respect for a relevant number of dogs. A more recent evaluation from 2019 [13] concluded that veterinarians under-report adverse events associated with NSAIDs and that owners should be better informed about the risks associated with long-term NSAID use in their dogs. Other, less frequently used medications for canine OA patients also appear to have limitations. Miles et al. (2020) found in their study that both tramadol and gabapentin can improve weight bearing in dogs diagnosed with OA and appear to be safe for short-term use in older patients, but concluded that the relatively high incidence of undesirable effects in some patients may outweigh the benefits [14]. Another substance, bedinvetmab, is a canine immunoglobulin G2 (IgG2) monoclonal antibody (mAb) intended to alleviate pain in OA patients. A study conducted by the manufacturer showed significant beneficial effects when CBPI was used in a treatment and placebo group [15]. However, Kronenberger (2023) evaluated the existing studies on this drug and concluded that they provide only inconclusive evidence of short-term safety and long-term efficacy [16]. Pacher et al. (2020) saw some

benefits from cannabidiol-rich hemp oil extract based on reduced pain scores, improvements in mobility and quality of life as defined by dog owners in a non-placebo-controlled field study [17]. Verrico et al. (2020) demonstrated the short-term efficacy of cannabidiol on quality of life as assessed by both owners and veterinarians in a double-blind, placebo-controlled study in dogs with spontaneous OA [18].

In addition to purely treating pain and alleviating symptoms without affecting the progressive nature of OA, several drugs have been investigated as potential structure-modifying OA drugs (SMOADs). SMOADs include numerous nutraceuticals, defined as supplementary feed items that serve a specific preventive or curative purpose (between "nutrition" and "pharmaceuticals") and are considered safe for consumption without side effects [19, 20]. SMOADs such as polysulfated glycosaminoglycans or hyaluronic acid aim to delay disease progression or repair OA lesions in affected joints rather than just alleviate the symptoms [21, 22]. However, no evidence from *in vivo* trials shows that these drugs can achieve these goals [23]. Studies investigating the effect of nutraceuticals such as green-lipped mussel extract, chondroitin sulfate, curcumin or blackcurrant leaf extract on OA in pets show promising effects in some cases but also provide contradictory or unsatisfactory results in clinical trials that lack consistent statistical significance or have an inadequate study design [24–29]. Omega-3 fatty acids (n3FA) with a defined concentration of eicosapentaenoic acid (EPA) in combination with a defined dose of vitamin E are the only substance group listed in EU Regulation 2020/3544 to support joint metabolism in OA in dogs and cats. Roush et al. (2010) reported that supplementation with n3FA from fish oil improved weight bearing, at least in the short term, as measured in a force-plate study in dogs with osteoarthritis [30]. Fritsch et al. (2010) found that n3FA fish oil effectively reduced pain medication in dogs with OA [31].

Specific bioactive collagen peptides (BCP) are another type of nutraceutical investigated *in vitro* and in vivo in several species. The mode of action of BCP supplementation complements the alleviating effects of other nutraceuticals by structurally improving affected tissue and slowing disease progression. Collagen peptides are characterized by a low molecular weight and high content of proline and hydroxyproline, resulting in strong resistance to intestinal digestion, increased transport efficiency through the intestinal barrier and consequently, enhanced bioavailability [32–34]. Collagen-derived peptides have been shown to accumulate in cartilage tissue, where they can stimulate chondrocytes to synthesize extracellular cartilage matrix molecules and counteract progressive tissue degeneration [35–39]. *In vitro* studies with canine chondrocytes showed that the biosynthesis of cartilage matrix molecules increased significantly after exposure to specific collagen peptides. In addition, a significant down-regulation of inflammatory cytokines and degenerative matrix metalloproteases indicated that the cartilage metabolism of canine chondrocytes is directly influenced by BCP [40]. Beneficial effects of BCP have also been demonstrated in horses with OA [41]. McAlindon et al. (2011) confirmed a positive effect of orally administered specific BCP on joint cartilage in human patients with mild OA, as measured by the diagnostic gold standard of magnetic resonance imaging [42]. In a controlled, randomized study of young, physically active adults, BCP supplementation also effectively reduced activity-related knee joint discomfort [43]. Thus, oral BCP is a promising nutraceutical for joint health in humans and dogs. Investigating the postulated positive effects in dogs can help to establish this substance in veterinary and human medicine and optimize the treatment of OA. As previously mentioned, animal models are routinely used for OA research, with dogs being one of the established and recommended models [5, 44–48], partly due to their greater anatomical similarity to humans including articular cartilage thickness [49]. They are also valued as a nearly ideal species for the translational study of human OA biomarkers [50]. Since dogs with OA can serve as a model for degenerative joint disease in humans, the BCP effects are likely to be transferable to human medicine.

Reviews of nutritional treatment protocols in dogs with chronic OA found that published studies vary in design and quality, ultimately leading to a lack of systematically developed evidence for optimal nutritional management of OA in dogs [28, 51].This may be due, at least in part, to the methodology used in some of the animal trials. Meaningful, valid and reliable measures of pain treatment are challenging even in self-evaluation studies in humans, in part due to the complex, multidimensional, sensory and emotional nature of pain [52]. In particular, when using questionnaires, it must be anticipated that placebo effects of caregivers may distort the results in studies with companion animals [53]. In this context, a "desirability bias" must also be considered [54], all the more so because of the owner-patient axis. A study design should therefore aim to include not only a placebo group but objective methods to test the efficacy of different preparations with postulated positive effects on joint cartilage.

In this double-blind, placebo-controlled field study named CANIS (Canine Arthrosis Nutritional Intervention Study), we investigated the effects of a 12-week supplementation with specific BCP in contrast to a combination of n3FA and vitamin E on the gait and activity of client-owned dogs with spontaneously occurring OA while also assessing pain and the perceived quality of life by owners. The aim was to gain valuable information—also for human OA patients—by applying a systematic study design in a translational canine model.

## Methods

All procedures and protocols were conducted in accordance with the guidelines of the Protection of Animals Act and the study was approved in written format by the responsible committee for animal welfare of the Veterinary Faculty, Ludwig-Maximilians-University, Munich (reference number 192-11-11-2019) between 2017 and 2021.

### Dietary supplements

Three different supplements (Table 1) were used in this trial: i) Commercially available Bioactive Collagen Peptides (PETAGILE®, GELITA AG Eberbach, Germany). These are derived from a special hydrolysis of predominantly type I porcine collagen, resulting in a mean molecular weight of approximately 6 kDa. They are characterized by the molecular weight fraction and amino acid sequence. BCPs have a high safety profile and are classified as a safe food by the European Food Safety Authority [55]. ii) A supplement named n3FA containing 19.5% docosahexaenoic acid (DHA), 12.6% EPA (both from Denk Ingredients, BIOMEGA Tech F 18:12 PVS), and vitamin E (Vitamin E 50% CWD, Nutrilo GmbH, Cuxhaven, Germany), selected as the only option under Commission Regulation (EU) 2020/354 [56] for the "support of joint metabolism in osteoarthritis" nutritional purpose. iii) The supplement without active ingredients contained cellulose (feed cellulose, Phrikolat Drilling Specialties GmbH, Hennef, Germany) combined with maltodextrin (C*Dry, Cargill Haubourdin SAS, Haubourdin,

**Table 1. Composition [%] of the three supplements used in this study.**

| Ingredients/supplements | BCP supplement | n3FA supplement | PLA supplement |
|---|---|---|---|
| Biomega Tech F 18:12 PVS (19.5% EPA, 12.6% DHA in 65% of the product, 35% silica gel) | - | 97.5% | - |
| Vitamin E acetate 50% CWD | - | 1.04% | - |
| Fish aroma | 3.8% | 1.46% | 3.8% |
| PETAGILE®, GELITA AG Eberbach, Germany | 96.2% | - | - |
| Maltodextrin C*Dry | - | - | 48.1% |
| Feed cellulose | - | - | 48.1% |

France) and was used as a placebo (PLA). All three supplements were mixed with fish aroma (3.8% in PLA and BCP, 1.4% in n3FA, Symrise AG, Holzminden, Germany) to mask the natural odor of the fish-derived n3FA product and to make them identical in appearance. All supplements consisted of a fine white powder and were packaged in ready-to-use pouches labeled only with a code and feeding instructions. The identical appearance of the content allowed for double-blind use, i.e., neither the supervising veterinarians nor the dog owners knew which supplement each dog received at the trial. The supplements were dosed according to body weight (BW). The BCP dosage was based on studies by Oesser et al. (2007) in mice [57] and results from field studies in dogs [40, 58, 59] with 240 ± 95mg BCP/kg BW. In the n3FA group, the dogs received 700 ± 115mg n3FA/kg BW combined with ≥ 2mg of vitamin E/kg BW, following regulation EG 2020/354 recommendations [56]. Dogs in the PLA group received 240 ± 50mg of the cellulose mixture per kg BW.The trial duration was 12 weeks, with daily oral administration of the respective supplement together with the dog's regular meal (white powder mixed with water and stirred into the regular feed). The patients were allocated to the coded and therefore blinded supplements, i.e. study groups, according to the order in which they entered the study. The study unblinding occurred after completion of the statistical evaluation of all results.

## Patients

Forty-one privately owned dogs previously diagnosed with chronic OA (≥ 3 months) by a veterinarian, without assessment of any primary causes, were randomly assigned to the BCP, n3FA or PLA group. Adult dogs (> 1 year) weighing between 10 and 60 kg were eligible for the study if they were not diagnosed with any other diseases requiring pain medication in addition to OA. Medications such as analgesics, corticosteroids or nutraceuticals for orthopedic applications had to be discontinued at least 6 weeks prior to enrolment to allow for participation in the study. Participation was not possible in cases where the medication could not be discontinued due to the severity of the condition. During the study period, pain medication was allowed for up to two consecutive days for acute and severe pain of any cause ("emergency medication"). Using analgesics for more than 2 days led to exclusion from the study. The owners were asked not to change the dog's diet during the study period. The data of all dogs that completed the study are listed in Table 2.

## Gait analyses

Gait analysis data was recorded at baseline (T0) and after 12 weeks of daily supplementation (T12) to assess the severity of OA symptoms based on weight-bearing parameters. The dogs were examined on a horizontal treadmill embedded in a platform with two separate belts and four integrated piezoelectric force plates in an identical manner at both T0 and T12. The ground reaction forces (GRF) of all four limbs were recorded simultaneously during walking. The speed was controlled by software in steps of 0.02 m/s (Simi Reality Motion Systems GmbH). The velocity was adjusted individually for each dog and gait and kept constant at T0 and T12. At least seven valid steps per limb and evaluation were analyzed using data collection software (Vicon Nexus, Vicon Motion Systems, Ltd., Oxford, United Kingdom; QuadruPedLocomotion, in-house software of Ludwig-Maximilians-Universität München). The parameters peak vertical force (PVF, % BW), vertical impulse (VI, % BW * s) and stance duration (DSP, % of step) of the valid steps of each limb and dog were calculated and summed as average values at T0 and T12. The limb affected by OA was defined at T0 using these parameters together with information from each dog's medical history. Only dogs with abnormal gait parameters of at least one limb at T0 in combination with the veterinary diagnosis of OA were

**Table 2. Breed, age, gender, and body weight of dogs with osteoarthritis participating in the study (mean ± SD).**

| | BCP | n3FA | PLA |
|---|---|---|---|
| | n = 11 | n = 11 | n = 9 |
| T0 body weight [kg] | 35 ± 15 | 26 ± 6.2 | 26 ± 8.5 |
| T12 body weight [kg] | 35 ± 15 | 25 ± 5.9 | 26 ± 8.6 |
| T0 age [years] | 7.9 ± 3.1 | 6.3 ± 4.0 | 6.1 ± 3.7 |
| Breeds | • Mixed breed, fn<br>• Mixed breed, mn<br>• Mixed breed, mn<br>• Mixed breed, fn<br>• Bernese Mountain Dog, mn<br>• Airedale Terrier, f<br>• Rottweiler, mn<br>• Border Collie, fn<br>• Pitbull Terrier, mn<br>• Hovawart, mn<br>• Rhodesian Ridgeback, mn | • Mixed breed, mn<br>• Mixed breed, mn<br>• Mixed breed, fn<br>• Mixed breed, fn<br>• Mixed breed, fn<br>• Mixed breed, fn<br>• Labrador Retriever, fn<br>• Golden Retriever, fn<br>• Gordon Setter, mn<br>• Bernese Mountain Dog, f<br>• Greyhound, f | • Mixed breed, fn<br>• Mixed breed, fn<br>• Mixed breed, fn<br>• Mixed breed, m<br>• Mixed breed, mn<br>• Mixed breed, mn<br>• German Shepherd, fn<br>• Australian Shepherd, fn<br>• Airedale Terrier, f |

BCP: Bioactive Collagen Peptides; n3FA: omega 3 fatty acids and vitamin E; PLA: placebo; T0: Start of the study period; T12: End of the study period; f = female, m = male, n = neutered; Body weight and age given as mean ± standard deviation

eligible for the study. For all averaged parameters (PVF, VI, DSP), the absolute (Δ) and relative difference (RPD) between T12 and T0 was calculated for both the affected and the contralateral limb. In addition, a symmetry index of PVF and VI (SIPVF, SIVI) was calculated for the affected limb pair as reported elsewhere [60]:

$$SI = ABS \left( 200 * \frac{PVFcontra - PVFaff.}{PVFcontra + PVFaff} \right)$$

with ABS = absolute amount, PVFcontra = PVF contralateral fore- or hindlimb, and PVFaff = PVF affected fore or hindlimb.

## Questionnaire

The owners were asked to complete a questionnaire on their perception of the dog's symptoms at T0 and T12. It included a previously validated pain assessment tool, the validated Canine Brief Pain Inventory (CBPI) [61]. The CBPI provides a summed score of ten questions on pain interference with specific movements, quality of life (QoL) and the estimated total pain on a visual analog scale (VAS; continuous slide control from 0–10 cm with 0 = no pain and 10 = most extreme pain). The relative change in CBPI scores between T12 and T0 was used to quantify the supplementation effects.

## Accelerometry

The dogs wore an accelerometer on a special collar (Actical©, Respironics, Phillips Healthcare, Amsterdam, The Netherlands) for two weeks prior to the T0 and T12 examination. The Actical© data was read and processed using the accompanying software (Actical™ software, version 3.10.0001, Koninklijke Philips Electronics N.V.). The total daily activity data was averaged over the complete days of the tracking period (phases of incomplete recording, e.g., on the first and last day of wearing the tracker, were identified and excluded from the analysis). The relative difference between each dog's T12 and T0 values was calculated to compare the groups. All participants, i.e., investigators and dog owners, were only unblinded once all data was available and the statistical analysis completed.

## Statistical evaluation

A power based sample size calculation was performed using the software G*Power 3.197 [62]. The statistical analysis was performed using IBM® SPSS® Statistics software version 29.0. As the Shapiro-Wilk test showed, the various metric parameters were not normally distributed. Therefore, only statistical measures were used without assuming a normal distribution. First, the data were summarized by calculating the mean, standard deviation, median and range (min and max). Box and whisker plots were used to present the data graphically (outliers were marked as circles). To statistically compare the data of the affected limb with the contralateral limb, we used the generalized linear model (GLM) in the repeated measurements variation (= generalized estimating equations—GEE). Within the GLM, the distribution of the dependent variable was specified as ordinal or gamma (only positive values in the case of metric data), and log was used as the link function. To analyze the effect of supplementation, the absolute difference ($\Delta$) and relative difference (RPD; in %) between T12 and T0 were calculated for each parameter. Within the groups, the results of the affected limb and the contralateral limb were analyzed for significant differences using GEE. The GLM was used to compare the SIPVF and SIVI of the three groups. In addition, the absolute and relative differences between the affected and unaffected limbs within each group were tested against zero (= no effect) using the One-Sample Wilcoxon Signed Rank Test. The symmetry indices were also tested with the One-Sample Wilcoxon Signed Rank Test against a cut-off of 9 (SIPVF) or 10 (SIVI), according to Wagmeister et al. (2021) [60]. The Kruskal-Wallis test with Bonferroni post hoc pairwise comparison was chosen to analyze the owners' questionnaire data. To analyze the activity data, the results for each dog were first categorized as decrease, increase or no change in activity level after 12 weeks of supplementation. Cross tabulation and Somers' D for ordinal data were used to analyze these categorical data. A p-value < 0.05 was considered significant. After the formal analysis of all results, the allocation of the dogs to the three study groups was unblended.

## Results

Of the dogs enrolled in the study, 10 were excluded for the following reasons: skin problems (1 x PLA; 1 x BCP); gastrointestinal problems (2 x n3FA); malignant tumor requiring pain medication (1 x BCP); unknown (2 x PLA; 1 x BCP; 2 x n3FA). Thirty-one dogs completed the study (Table 2).

The age of the dogs at T0 did not differ significantly between the supplementation groups (p = 0.747). The starting and end weights also showed no significant differences between the groups (p = 0.413 and p = 0.094, respectively). The acceptance of all supplements was sufficient; the dogs fully consumed the supplements added to the regular diet.

### Gait

Before the start of the three-month supplementation phase (T0), the gait analysis parameters PVF, VI and DSP showed lower values for the OA-affected limbs than for the unaffected contralateral limbs (Table 3). The eligibility of the thirty-one dogs was confirmed based on these results.

As expected, there were no significant differences between T0 and T12 for all three parameters in the unaffected limbs (S1 and S2 Tables). Therefore, only the affected limbs are detailed in the following presentation of the results: In the PLA group, there was no significant effect on PVF (Fig 1; p = 0.809), VI (Fig 2; p = 0.578) and DSP (p = 0.946) as measured by $\Delta$, i.e., the differences between T0 and T12. The $\Delta$ calculated for the dogs in group n3FA showed no significant difference between the two time points: the measured values for PVF and DSP at T0

**Table 3. Comparison of the gait parameters peak vertical force (PVF), vertical impulse (VI) and duration of stance phase (DSP) between the affected and unaffected limbs before the start of the supplementation phase (T0).** Due to the different weight deposition, the data of the forelimbs and hind limbs were analyzed separately.

| Parameter | forelimb | | | hind limb | | |
|---|---|---|---|---|---|---|
| | affected | unaffected | p | affected | unaffected | p |
| | (n = 14) | (n = 48) | | (n = 18) | (n = 44) | |
| PVF | 54.05 | 58.4 | 0.018 | 38.35 | 39.95 | <0.001 |
| | [48; 69.4] | [50.7; 75.3] | | [24.4; 42.8] | [33.3; 50.6] | |
| VI | 18.35 | 20.45 | <0.001 | 12.44 | 11.80 | 0.390 |
| | [15.6; 22.3] | [16.2; 26.9] | | [7.8; 21.2] | [7.8;21.2] | |
| DSP | 69 | 70.38 | 0.020 | 67.5 | 67.63 | 0.668 |
| | [64.5; 73.8] | [65.5; 75.5] | | [57.5; 74.5] | [59.3; 75.8] | |

Data are presented as median [min; max], and p-values < 0.05 were considered significant.

and T12 were very similar (p = 0.972). In this group, VI increased slightly but was not statistically significant (p = 0.273). On the other hand, there was a significant increase in Δ for PVF in the BCP group compared to the unaffected limbs (p = 0.015). BCP supplementation resulted in a significant increase in Δ VI from T0 to T12 (p = 0.004), while Δ DSP increased only numerically (p = 0.312).

When comparing the RPD percentages in the gait parameters of the affected and the unaffected limbs, the PLA group showed a decrease, i.e., clinical deterioration, in all three parameters measured in the affected limb during the 12-week period, which was significant for PVF (Fig 3), close to significance for VI (p = 0.072), or not significant (DSP). In the n3FA group, there was a trend towards statistical significance concerning the increase in PVF (p = 0.163) but no changes in RPD for VI (p = 0.703) and DSP (p = 0.617). On the other hand, there was a

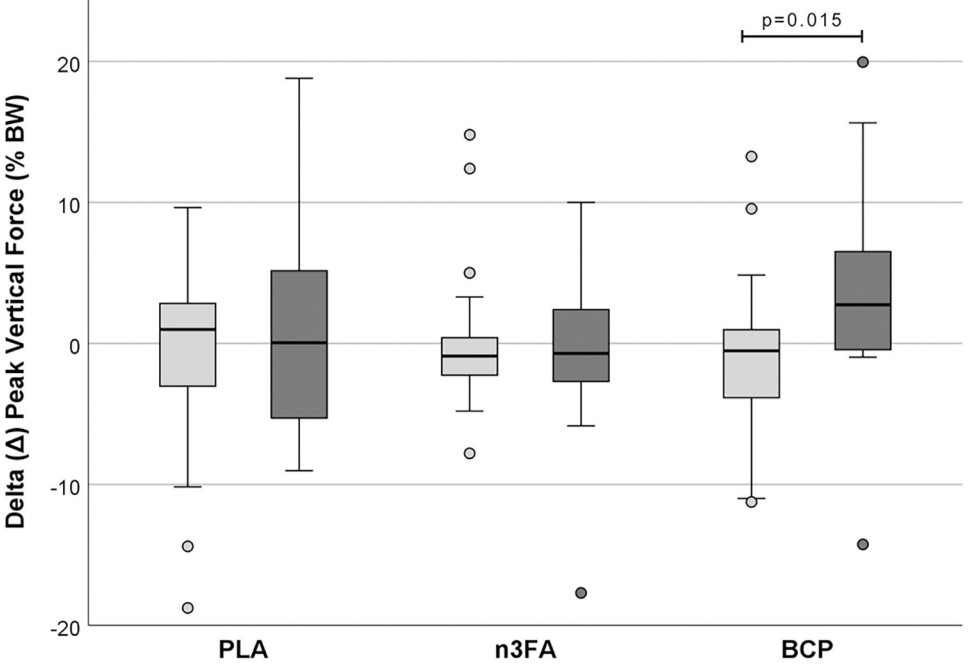

**Fig 1. Difference in peak vertical force (Δ PVF) between T12 and T0 [% BW].** The light gray boxes represent the unaffected limbs, the darker boxes the affected limbs.

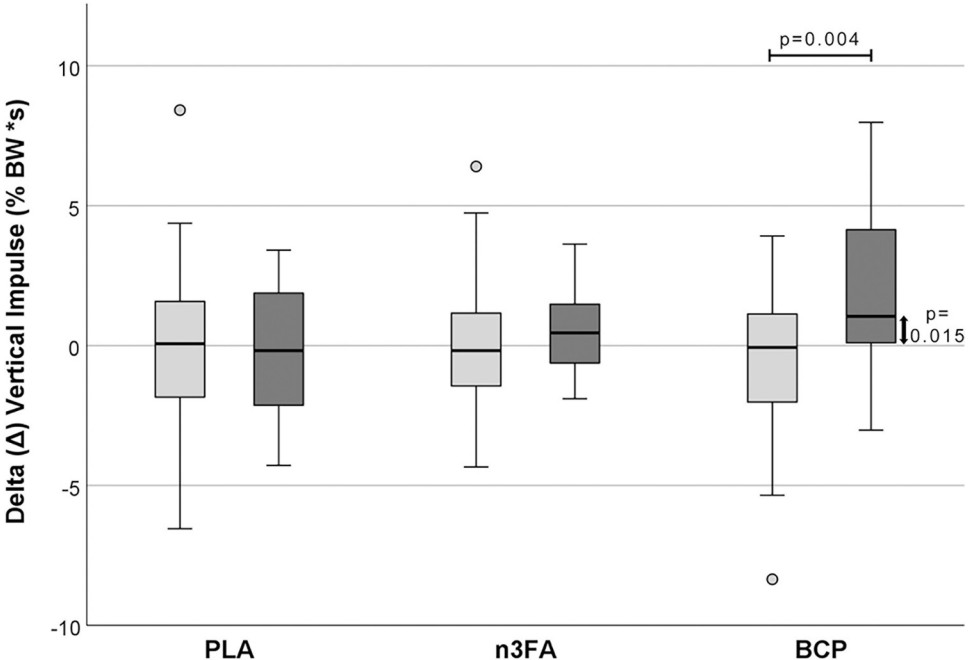

**Fig 2. Difference in vertical impulse (Δ VI) between T12 and T0 [% BW*s].** The light gray boxes represent the unaffected limbs, the darker boxes the affected limbs.

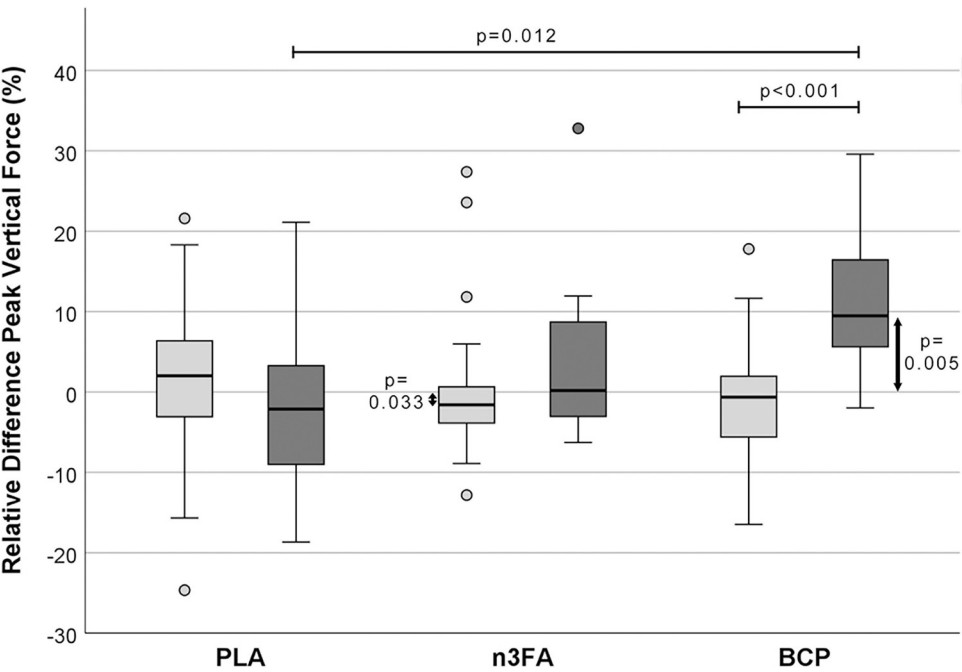

**Fig 3. Relative difference of peak vertical force (PVF) between T12 and T0 [%].** The light gray boxes represent the unaffected limbs, the darker boxes the affected limbs.

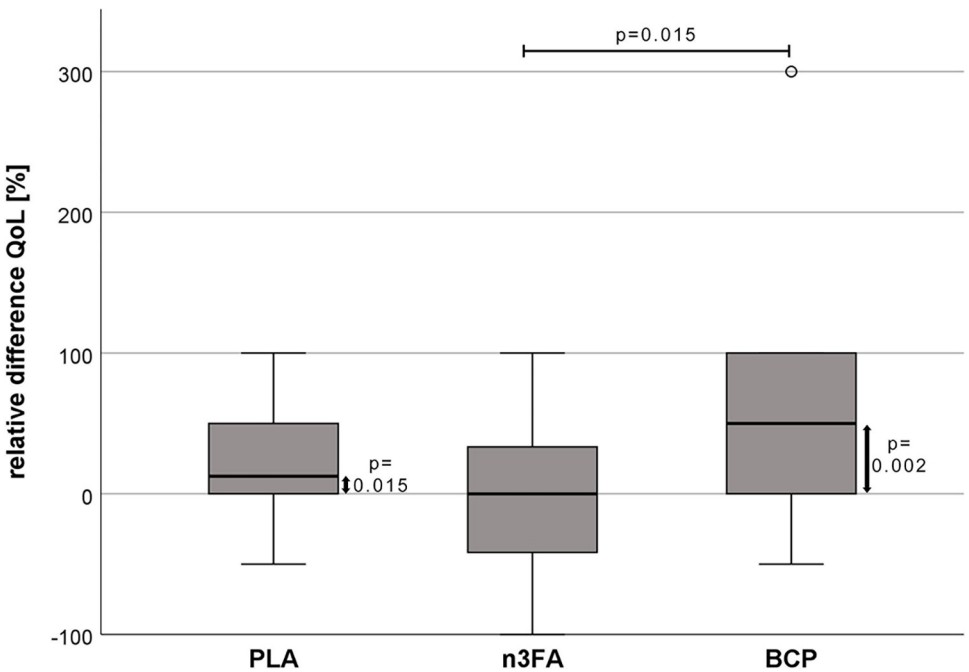

**Fig 4. Relative difference in the quality of life (QoL) [%] between T0 and T12.**

highly significant improvement in PVF in the BCP group ($p < 0.001$), an effect that was significant compared to the PLA group ($p = 0.012$) or the unaffected limb of the same group ($p < 0.001$). In the BCP group, the treadmill examination also showed a trend towards an increase in VI ($p = 0.157$) but no effect for DSP ($p = 0.643$).

SIPVF and SIVI (S3 Table) were tested for their threshold values of 10% and 9%, respectively. A symmetry index above this value indicates clinically apparent lameness with differing degrees of weight loading on the contralateral limbs [60]. No changes in SI PVF and SI VI were observed after 12 weeks of supplementation in the PLA group. The SI PVF showed a tendency to decrease in the n3FA group ($p = 0.117$), while the improvement in the BCP group was significant ($p = 0.041$). The SI VI remained unaffected by the supplements.

## Questionnaire

The questionnaire results show a significant improvement in QoL in the BCP group (relative difference: $54.55 \pm 73.32\%$) when the results of the validated CBPI score are compared with the n3FA group ($-2.54 \pm 62.13\%$; $p = 0.015$; Fig 4). The relative difference in QoL was also numerically higher in BCP compared to PLA ($25.00 \pm 40.66\%$; $p = 0.517$). When tested for the zero-change hypothesis, both BCP ($p = 0.002$) and PLA ($p = 0.015$) supplementation significantly improved QoL. The pain score, calculated by adding the results of the first ten CBPI questions and the VAS, showed no systematic differences between the treatment groups.

## Accelerometry

The use of accelerometers at the beginning and end of the 12-week period of supplementation of the test substances and the placebo had no significant effect on activity, neither in relation to the time of measurement nor the type of supplementation (Somers' d = -0.137; $p = 0.244$).

## Discussion

Various studies and literature reviews report that oral supplementation with collagen hydrolytes can be effective in mitigating the clinical sings of OA but also conclude that further systematic studies are needed [40, 58, 63–66]. In the present field study, a 12-week oral supplementation of specifically hydrolyzed collagen, or BCP, was tested against the effect of another active substance (n3FA plus vitamin E) and a placebo in dogs with naturally occurring and diagnosed OA. The dogs' gait was examined on a horizontal treadmill with integrated piezoelectric force plates. In addition, the validated CBPI, determined by the dog owners at the beginning and end of the study using a questionnaire, and activity measurements with accelerometers were evaluated.

Clearly defined, blinded and placebo-controlled systematic studies are needed to assess possible beneficial effects of the nutritional supplement BCP. The use of canine patients diagnosed with OA in research, applying clearly defined inclusion and exclusion criteria as well as maximally objective assessment methods, may be valuable as a translational model for human OA patients. In this study, dogs with OA diagnosed by a veterinarian and with abnormal gait parameters of at least one limb at T0 were included. On the other hand, participation was not possible in cases where the medication could not be discontinued for at least 4.5 months (study duration plus wash out period before its start) due to the severity of the condition. Only dogs which did not depend on medication or nutraceuticals were eligible for the study. Consequently, the severity of OA can be categorized as mild to moderate.

### Gait

The force plate measurements on a treadmill were chosen as a more objective method than questionnaires for owners [67–69]. They showed partially significant improvements in the evaluated gait analysis parameters VI, PVF, and symmetry indices as kinetic measures, and DSP, determined in the affected and contralateral limbs in dogs after 12 weeks of oral BCP intake. A positive value for the difference Δ between T12 and T0 indicates an improvement of the parameter at T12, while negative values indicate a deterioration of the respective parameter. In the unaffected limbs of all three supplementation groups, the difference for all gait parameters was close to zero and showed no significant change between T0 and T12. The VI was significantly increased only in the affected limb of the BCP group, indicating that the affected leg was more heavily loaded after three months of BCP supplementation. The improvement in applied pressure, measured as the relative difference in PVF, was only significant in the affected limb of the dogs in the BCP group. The increase between T0 and T12 was highly significant. This development differed significantly from that in the PLA placebo group. There was no change in the DSP in either the PLA or n3FA group, while the parameter showed a numerical increase in the BCP group. In the PLA group, all measured parameters tended to deteriorate, as is to be expected in a progressive disease such as OA after three months. This trend was not observed in the n3FA group, where no systematic differences were found between the two time points. This suggests that the combination of effective substances approved by EU legislation to support joint metabolism in OA in dogs and cats, appears to have slowed disease progression in this study.

Orally administered collagen hydrolysates are absorbed (humans: [70]; dogs: [71]) and can be detected in tissues such as cartilage after administration of the labeled product [72]. The pronounced resistance to intestinal digestion, high transport efficiency and consequently high bioavailability is associated with the low molecular weight and high proportion of proline and hydroxyproline in collagen peptides [32–34]. The two main modes of action of orally administered collagen peptides are the stimulation of extracellular matrix molecules synthesis, especially proteoglycans, and the down-regulation of inflammatory cytokines and degenerative matrix metalloproteases. This suggests that optimal efficacy of collagen peptide administration

may require that a certain degree of OA is not exceeded in the affected joint(s) of the patient. Early intervention, if not preventive administration of functional collagen peptides such as BCP in predisposed or potentially affected individuals (chondrodystrophy, trauma, increased stress from sport, overweight and obesity etc.) is therefore recommended to maintain joint health and functionality for as long as possible, as also suggested by Kwatra (2020) [73]. This is increasingly important in the aging pet and human population with a high prevalence of overweight and obesity [74, 75].

It is important to note that collagen hydrolytes are not a homogenous group of peptides. The composition and characteristics, and thus the mode of action and efficacy as an oral therapeutic agent for OA, are influenced by the tissue from which the material is derived and by the processing steps such as hydrolysis method and time, pH, temperature and filtration, resulting in differences in the molecular size and amino acid pattern of the product [76–79]. Therefore, the results of one study are not necessarily transferable, especially if different active substances, i.e., differently produced collagen hydrolytes, are used.

In this study, examination of the gait of dogs with naturally occurring OA on a treadmill equipped with force plates clearly showed positive effects of BCP, which significantly mitigated and alleviated clinical signs of the progressive disease. This safe oral nutritional supplement has impressively shown properties superior to the only nutrient combination approved by European legislation for such use. This systematic, double-blind, placebo-controlled study in dogs with naturally occurring OA diagnosed by veterinarians, using the objective parameter of weight-bearing measuring, demonstrated the superior efficacy of BCP supplementation to the approved combination of omega-3-fatty acids with vitamin E in a clinical setting. This type of study is necessary to translate the results of *in vitro* studies and studies in small animal models with induced OA into practically relevant applications.

## Questionnaire

After 12 weeks of supplementation, QoL was higher in the PLA group and more pronounced in the BCP group. In the n3FA group, no changes were noted by the patients' caregivers. The improvement in QoL, as assessed by the owners using the validated CBPI, was significantly higher in the BCP group than the n3FA group and numerically higher than in the PLA group. No significant differences were found in pain scores on the CBPI. A worsening of clinical signs such as pain, would have been conceivable after the three-month trial period. However, this was not observed in any of the three groups. Even though the method of asking owners about their perception of signs in their dogs is known to be subjective and possibly imprecise [80], the clear improvement in QoL in the BCP group is remarkable. The CBPI pain interference score was found not to be associated with owner response bias and is therefore recommended as a clinical outcome measure for chronic pain and pain-related disability in dogs with OA [81]. Thus, even if a certain placebo effect has to be considered, as detected in the PLA group, the improvement was significantly more pronounced in the BCP group. The fact that the more objective treadmill measurements also showed the most significant overall improvement in the patients supplemented with BCP supports the questionnaire's positive results. However, the relatively short supplementation period may not have resulted in a noticeable improvement in all clinical signs typically seen in OA. Further longitudinal studies examining the progression of OA and the beneficial effects of BCP on the cartilage are warranted.

## Accelerometry

The measurement of daily activity at the beginning and end of the supplementation period using accelerometers showed no significant differences between the groups. The method itself

is known to be demanding and dependent on the system used and the frequency of read-outs but is nonetheless recommended [82]. One of the reasons for the overall low statistical effect for this parameter is the high degree of individual variation at both time points, independent of the group. It has also to be taken into account, that in most canine OA patients, free activity is likely to be limited in duration and intensity by the owner. In particular, older dogs with known medical conditions may be walked for shorter periods and on a leash instead of being allowed to run free. In addition, the owners were asked not to change their dog's exercise schedule too much during the study so that we could detect changes in spontaneous or voluntary activity. Thus, the dogs may not have had the opportunity to engage in more voluntary activity because the owners set the time for the walks. Therefore, it is likely that accelerometry could not detect more subtle effects in client-owned dogs in the limited time frame of the study.

## Conclusion

Oral BCP supplementation over a relatively short period of 12 weeks can significantly improve the clinical signs of naturally occurring OA in dogs. The effect of BCP was significantly greater than that of n3FA and vitamin E, which is currently the only nutritional strategy legally approved to improve OA in dogs. This is proof of the efficacy of the non-invasive and safe treatment option with BCP in dogs and should also focus attention on the use of oral BCP supplementation in feline and especially human OA patients. The present study showed impressively that the clinical signs of naturally occurring canine OA were not only stopped but improved by using objective methods after 12 weeks of oral BCP supplementation. Given the benefits and safety of BCP supplementation as well as the lack of known undesirable effects, early onset, long-term treatment with BCP in human, pet and equine OA patients seems advisable. However, longitudinal studies and trials investigating the possibility of delaying or even preventing the onset of OA by oral BCP supplementation are needed.

## Supporting information

**S1 Table. Absolute difference (Δ) in gait analysis parameters of affected and unaffected limbs between the second (T12) and first (T0) examination, grouped by supplementation (absolute difference; median [min; max]).**
(DOCX)

**S2 Table. Relative difference [%] in gait analysis parameters of affected and unaffected limbs between the first (T0) and second (T12) examination (%; median [min; max]).**
(DOCX)

**S3 Table. Probability of error (p) for the symmetry indices of peak vertical force and vertical impulse concerning the threshold value of 10 (SI PVF) and 9 (SI VI), respectively.**
(DOCX)

## Acknowledgments

We thank all participants who volunteered for this study.

## Author Contributions

**Conceptualization:** Britta Dobenecker, Jutta Hugenberg.

**Formal analysis:** Britta Dobenecker, Linda Franziska Böswald, Sven Reese.

**Investigation:** Britta Dobenecker, Linda Franziska Böswald, Stephanie Steigmeier-Raith, Lukas Trillig.

**Methodology:** Britta Dobenecker, Linda Franziska Böswald, Jutta Hugenberg.

**Resources:** Steffen Oesser, Andrea Meyer-Lindenberg, Jutta Hugenberg.

**Supervision:** Britta Dobenecker.

**Visualization:** Britta Dobenecker, Sven Reese.

**Writing – original draft:** Britta Dobenecker.

**Writing – review & editing:** Linda Franziska Böswald, Sven Reese, Stephanie Steigmeier-Raith, Lukas Trillig, Steffen Oesser, Michael Schunck, Andrea Meyer-Lindenberg, Jutta Hugenberg.

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
