## [Decision Letter · Decision Letter 0]

23 Apr 2024

PONE-D-24-10151The oral intake of specific Bioactive Collagen Peptides (BCP) improves gait and quality of life in canine osteoarthritis patients - a translational large animal model for a nutritional therapy optionPLOS ONE

Dear Dr. Dobenecker,

Thank you for submitting your manuscript to PLOS ONE. After careful consideration, we feel that it has merit but does not fully meet PLOS ONE’s publication criteria as it currently stands. Therefore, we invite you to submit a revised version of the manuscript that addresses the points raised during the review process.

**Dear Authors,****please provide all **additional** information.****with best regards****Ewa Tomaszewska**

We look forward to receiving your revised manuscript.

Kind regards,

Ewa Tomaszewska, DVM Ph.D

Academic Editor

PLOS ONE

Journal Requirements:

   "CRI, Collagen Research Institute GmbH, Kiel, Germany.

Supplements/placebo in this study were funded by GELITA AG, Germany."

   "Steffen Oesser is co-inventor of patents concerning the use of collagen peptides. Michael Schunck is employed by CRI, Kiel, Germany. Julia Hugenberg is employed by GELITA AG, Eberbach, Germany. At the time of investigation, Britta Dobenecker, Linda Böswald, Sven Reese, Stephanie Steigmeier-Raith, Lukas Trillig, and Andrea Meyer-Lindenberg were employed by LMU Munich, Germany, which received financial compensation for the study. Britta Dobenecker and Linda Böswald received compensation from CRI for medical writing, and Sven Reese for statistical analyses."

We note that one or more of the authors are employed by a commercial company: GELITA AG, Eberbach, Germany

6. Please include your tables as part of your main manuscript and remove the individual files. Please note that supplementary tables (should remain/ be uploaded) as separate "supporting information" files

Additional Editor Comments:

Dear Authors,

some issues should be clarify:

1. L170- the dog's regular meal. This should be presented in table with basal ingreadients.

2. L 175- Further information are needed about the dogs, such as their breed, sex, and age distribution, to understand the population's diversity.

3. Details about the severity of osteoarthritis among the patients, possibly categorized into mild, moderate, and severe.

4. Information on any past treatments or therapies for osteoarthritis and their outcomes.

5. Details about the specific diagnostic methods used to confirm OA and how severity was assessed.

6. Further clarification on the specific inclusion and exclusion criteria, including any health checks or requirements for participation.

7. More information on the study's design, including duration, controls, randomization methods, and blinding, to understand the study's structure.

8. Indications that the study complied with ethical guidelines and that appropriate consent was obtained from the dogs' owners.

9. Information about the follow-up process and how the researchers monitored the dogs' health and progress during and after the study.

10. the dosages, and frequency of administration of all suplemments should be included. It cannot be described as "with a minimum of...". who decided about the final dose? The treatment should be described precisely.

11. was the normal distribution of the data checked?

12. it should be described in detail how the dogs were assigned to these three groups, and what the dogs in the groups looked like - exactly age, weight, sex, degree of OA.

Reviewers' comments:

Reviewer's Responses to Questions

**Comments to the Author**

1. Is the manuscript technically sound, and do the data support the conclusions?

Reviewer #1: Yes

2. Has the statistical analysis been performed appropriately and rigorously? 

Reviewer #1: Yes

3. Have the authors made all data underlying the findings in their manuscript fully available?

Reviewer #1: Yes

4. Is the manuscript presented in an intelligible fashion and written in standard English?

Reviewer #1: Yes

5. Review Comments to the Author

Reviewer #1: The manuscript describes the primary results on oral supplementation of BCP in the treatment of OA in dogs. It is very usefull for veterinary clinicians. I have only a few questions before publication.

line 173: Have the authors excluded all locomotry disese, that may be misdiagnosed with OA ?

line 243-245: What did author mean: "unknown" - please explain.

6. PLOS authors have the option to publish the peer review history of their article (what does this mean?). If published, this will include your full peer review and any attached files.

Reviewer #1: No

---

## [Author Response · Author response to Decision Letter 0]

9 Jul 2024

also added as MS Word file (see attached files)

Dear reviewers

Thank you for your time and efforts invested in reviewing our manuscript. 

We amended the manuscript based on your comments and tried to implement all your suggestions thoroughly. Please find our response to your comments below.

Regarding the journal requirements:

1. The style was adapted to the requirements

2. Funding information was deleted from the text (funding statement only)

3. The funding section was reworded: Supplements/placebo in this study were funded by GELITA AG, Germany. The funder had no influence on study design, the collection, analysis and interpretation of data, the decision to publish, and the preparation of the manuscript. The funder [JH] provided support in the form of financial compensation for authors, but did not have any additional role in the study design, data collection and analysis, decision to publish, or preparation of the manuscript.

4. The conflict of interest section was reworded because the study was solely funded by GELITA AG. Conflict of interest: J Jutta Hugenberg is employed by GELITA AG, Eberbach, Germany. At the time of investigation, Britta Dobenecker, Linda Böswald, Sven Reese, Stephanie Steigmeier-Raith, Lukas Trillig, and Andrea Meyer-Lindenberg were employed by LMU Munich, Germany, which received financial compensation for the study. This does not alter our adherence to PLOS ONE policies on sharing data and materials.

5. In the methods section the following information was added: All procedures and protocols were conducted in accordance with the guidelines of the Protection of Animals Act and the study was approved in written format by the responsible committee for animal welfare of the Veterinary Faculty, Ludwig-Maximilians-University, Munich (reference number 192-11-11-2019). 

6. The tables 1 to 3 were included as part of the main manuscript 

1. L170- the dog's regular meal. This should be presented in table with basal ingredients.

Authors’ response: There was a huge variety of diet compositions that the enrolled patients received. It was made sure that no diet components containing nutraceuticals were fed 6 weeks prior to the start of the trial (see also line 181ff: Medications such as analgesics, corticosteroids or nutraceuticals for orthopedic applications had to be discontinued at least 6 weeks prior to enrolment to allow for participation in the study). It was decided to abstain from performing a complex and time-consuming ration calculation and diet evaluation for all dogs, especially because the feeding regime remained constant throughout the trial (the owners were informed that the diet of the dog should not be changed during the study phase). To underline this, the following sentence was added to the passage ‘patients’ in chapter ‘methods’ (line 187): The owners were asked not to change the dog’s diet during the study period.

2. L 175- Further information are needed about the dogs, such as their breed, sex, and age distribution, to understand the population's diversity.

Authors’ response: Table 2 gives the body weight (mean ± standard deviation), the age and lists the breeds of the canine patients in each group. We added the gender of the patients to this table.

3. Details about the severity of osteoarthritis among the patients, possibly categorized into mild, moderate, and severe.

Authors’ response: One important inclusion criterion was that the consequences of the OA, which had to be diagnosed by a vet, was measurable in the treadmill examination at T0 (see also the following sentence in the materials & methods section: Only dogs with abnormal gait parameters of at least one limb at T0 in combination with the veterinary diagnosis of OA were eligible for the study). Therefore, only dogs with clinical signs that showed during the treadmill measurements were eligible; this equates to the lower threshold for severity. On the other hand, participation was not possible in cases where the medication could not be discontinued for at least 4.5 months due to the severity of the condition. Only dogs which did not depend on medication or nutraceuticals were eligible for the study. This automatically excludes cases of more pronounced severity. The severity of AO in the study population can be overall categorized as mild to moderate).

This information was added to the discussion. 

Apart from this, a main problem here is that clinical signs and the degree of OA do not correlate well enough to rank the OA of canine patients. The same applies more or less to the treadmill /force plate measurements. The SI (symmetry index) probably gives the best estimate but this often does not suffice to rank correctly. Even evaluation of radiological methods can deviate depending on the applied scoring systems that often cannot be converted from the specific joint they are validated for. 

4. Information on any past treatments or therapies for osteoarthritis and their outcomes.

Authors’ response: This information was not specifically collected. It was made sure that any medication including the use of nutraceuticals was discontinued at least 6 weeks 

5. Details about the specific diagnostic methods used to confirm OA and how severity was assessed.

Authors’ response:

We accepted the veterinarian’s statement that OA was diagnosed in an individual. Regarding the severity of the OA please see the response to the comment #3. 

6. Further clarification on the specific inclusion and exclusion criteria, including any health checks or requirements for participation.

Authors’ response:

All inclusion and exclusion criteria are mentioned in the materials and methods section (body weight range, no other diseases requiring pain medication, medication such as analgesics and corticosteroids or their use for more than 2 days, respectively, use of nutraceuticals 6 weeks before or during the study, acceptance to participate in the study / treadmill measurements)

7. More information on the study's design, including duration, controls, randomization methods, and blinding, to understand the study's structure.

Authors’ response:

We have added the following information to the manuscript:

The study was conducted between 2017 and 2021.

A sentence was added to materials and methods explaining the randomization process:

The patients were allocated to the coded and therefore blinded supplements, i.e. study groups, according to the order in which they entered the study. 

Blinding: In the materials & methods section (dietary supplements) the following sentences explain the blinding during the study: All supplements consisted of a fine white powder and were packaged in ready-to-use pouches labeled only with a code and feeding instructions. The identical appearance allowed for double-blind use, i.e., neither the supervising veterinarians nor the dog owners knew which supplement each dog received at the trial.

We added the following information: The study unblinding occurred after completion of the statistical evaluation of all results (to materials & methods, dietary supplements). As well as: After the formal analysis of all results, the allocation of the dogs to the three study groups was unblended (materials & methods, statistical evaluation).

Further point:

The patients of the placebo group acted as a control.

8. Indications that the study complied with ethical guidelines and that appropriate consent was obtained from the dogs' owners.

Authors’ response: The following information was added at the beginning of the materials & methods section:

All procedures and protocols were conducted in accordance with the guidelines of the Protection of Animals Act and the study was approved by the representative of the Veterinary Faculty for animal welfare.

9. Information about the follow-up process and how the researchers monitored the dogs' health and progress during and after the study.

Authors’ response: 

A follow-up was not systematically done, just in single cases when the owners requested a supplement for further use. The dogs were not monitored during the study. The researchers were available in case of specific questions or incidents (e.g., exclusion of a dog in case of pain medication for more than 2 days etc.). 

10. the dosages, and frequency of administration of all suplemments should be included. It cannot be described as "with a minimum of...". who decided about the final dose? The treatment should be described precisely.

Authors’ response:

The owner were supplied with enough dietary supplement prepacked in pouches for the whole study period. The packages were stored refrigerated and added directly to wet food or diluted in a small amount of water and then added to a dry food on a daily basis. 

We adapted the information in the materials & methods section as follows:

The BCP dosage was based on studies by Oesser et al. (2007) in mice [57] and results from field studies in dogs [40,58,59] with 240 ± 95mg BCP/kg BW. In the n3FA group, the dogs received 700 ± 115mg n3FA/kg BW combined with ≥ 2mg of vitamin E/kg BW, following regulation EG 2020/354 recommendations [56]. Dogs in the PLA group received 240 ± 50mg of the cellulose mixture per kg BW.

11. was the normal distribution of the data checked?

Authors’ response:

Of course. A colleague specialized in statistics was involved in the statistical analysis of results.

12. it should be described in detail how the dogs were assigned to these three groups, and what the dogs in the groups looked like - exactly age, weight, sex, degree of OA.

Authors’ response: 

Details about the dogs enrolled in the study can be found in table 2. More details, e.g., the degree of OA, were added to the text (see answer to the issue #3).

---

## [Decision Letter · Decision Letter 1]

23 Jul 2024

The oral intake of specific Bioactive Collagen Peptides (BCP) improves gait and quality of life in canine osteoarthritis patients - a translational large animal model for a nutritional therapy option

PONE-D-24-10151R1

Dear Dr. Britta Dobenecker,

We’re pleased to inform you that your manuscript has been judged scientifically suitable for publication and will be formally accepted for publication once it meets all outstanding technical requirements.

Kind regards,

Ewa Tomaszewska, DVM Ph.D

Academic Editor

PLOS ONE

Additional Editor Comments (optional):

Reviewers' comments:

Reviewer's Responses to Questions

**Comments to the Author**

1. If the authors have adequately addressed your comments raised in a previous round of review and you feel that this manuscript is now acceptable for publication, you may indicate that here to bypass the “Comments to the Author” section, enter your conflict of interest statement in the “Confidential to Editor” section, and submit your "Accept" recommendation.

Reviewer #1: All comments have been addressed

2. Is the manuscript technically sound, and do the data support the conclusions?

Reviewer #1: Yes

3. Has the statistical analysis been performed appropriately and rigorously? 

Reviewer #1: Yes

4. Have the authors made all data underlying the findings in their manuscript fully available?

Reviewer #1: Yes

5. Is the manuscript presented in an intelligible fashion and written in standard English?

Reviewer #1: Yes

6. Review Comments to the Author

Reviewer #1: Tha authors made all indicated corrections. I accept the manuscript in present form and reccommend it for publications

7. PLOS authors have the option to publish the peer review history of their article (what does this mean?). If published, this will include your full peer review and any attached files.

Reviewer #1: No

---

## [Editor Report · Acceptance letter]

10 Sep 2024

PONE-D-24-10151R1 

PLOS ONE

Dear Dr. Dobenecker, 

I'm pleased to inform you that your manuscript has been deemed suitable for publication in PLOS ONE. Congratulations! Your manuscript is now being handed over to our production team.

Kind regards, 

on behalf of

Professor Ewa Tomaszewska 

Academic Editor

PLOS ONE